# HARDWARE-RESTRICTION-AWARE TRAINING (HRAT) FOR MEMRISTOR NEURAL NETWORKS

## ABSTRACT

Memristor neural network (MNN), which utilizes memristor crossbars for vector-matrix multiplication, has huge advantages in terms of scalability and energy efficiency for neuromorphic computing. MNN weights are usually trained offline and then deployed as memristor conductances through a sequence of programming voltage pulses. Although weight uncertainties caused by process variation have been addressed in variation-aware training algorithms, efficient design and training of MNNs have not been systematically explored to date. In this work, we propose Hardware-Restriction-Aware Training (HRAT), which takes into account various non-negligible limitations and non-idealities of memristor devices, circuits, and systems. HRAT considers MNN's realistic behavior and circuit restrictions during offline training, thereby bridging the gap between offline training and hardware deployment. HRAT uses a new batch normalization (BN) fusing strategy to align the distortion caused by hardware restrictions between offline training and hardware inference. This not only improves inference accuracy but also eliminates the need for dedicated circuitry for BN operations. Furthermore, most normal scale signals are limited in amplitude due to the restriction of non-destructive threshold voltage of memristors. To avoid input signal distortion of memristor crossbars, HRAT dynamically adjusts the input signal magnitude during training using a learned scale factor. These scale factors can be incorporated into the parameters of linear operation together with fused BN, so no additional signal scaling circuits are required. To evaluate the proposed HRAT methodology, FC-4 and LeNet-5 on MNIST are firstly trained by HRAT and then deployed in hardware. Hardware simulations match well with the offline HRAT results. We also carried out various experiments using VGG-16 on the CIFAR datasets. The study shows that HRAT leads to high-performance MNNs without device calibration or on-chip training, thus greatly facilitating commercial MNN deployment.

## 1 INTRODUCTION

Memristor neural network (MNN) has emerged as an increasingly feasible option to alleviate the scalability and energy efficiency challenges in neuromorphic computing. While several small-scale MNNs have been prototyped Li et al. (2018); Yao et al. (2020); Wan et al. (2022), efficient design and training of MNNs require an in-depth understanding of various restrictions from device, circuit, and system perspectives. These hardware restrictions include weight uncertainty noise caused by memristor variability and a limited number of programming pulse cycles to tune memristor conductance (*e.g.*, 500 in Yao et al. (2020)), weight quantization noise due to limited states of memristor conductance (*e.g.*, 5- and 4-bit in Yao et al. (2020); Wan et al. (2022)), non-destructive threshold voltage of memristors Jo et al. (2010), limited output swing of operational amplifiers Karki (2021), and bias quantization noise from finite-resolution digital-to-analog converters (DACs). These hardware restrictions collectively reduce the accuracy of MNN inference. Ignoring these hardware restrictions during software offline training may result in poor inference or even functional failure.

As a critical step in network training, batch normalization (BN) can accelerate training convergence Ioffe & Szegedy (2015). The scale and shift operations of BN can be merged into the previous linear operation (*e.g.*, fully connected or convolutional layer) after training. In this way, the hardware complexity and cost of MNNs are alleviated, as BN does not require explicit memristor crossbars in

on-chip deployment. We envision that the aforementioned hardware restrictions have a significant impact on BN fusion (also known as BN folding) in MNNs. For example, bias distortion caused by DACs and the limited output swing of operational amplifiers should be considered during BN fusion. Although several BN fusing strategies Jacob et al. (2018); Krishnamoorthi (2018); PyTorch (2022); Wan et al. (2022) have been reported for quantization-aware training, unfortunately, dedicated BN fusion strategies for MNN training and hardware deployment has so far received little attention. As a result, it is imperative to develop hardware-restriction-aware BN fusing strategies to align signal distortion caused by hardware restrictions before and after BN fusion in MNNs.

In this work, we propose a Hardware-Restriction-Aware Training (HRAT) method, which takes into account various non-negligible restrictions and non-idealities from device, circuit, and system perspectives. HRAT considers realistic behavior and hardware restrictions of MNNs during offline training, thereby bridging the gap between offline training and hardware deployment. The key contributions of this work are summarized as follows:

- We model various hardware restrictions of MNNs and integrate them into training, enabling hardware-restriction-aware training (HRAT). HRAT uses a new BN fusing strategy to align the restriction-induced distortion between offline training and hardware inference. This not only improves inference accuracy but also eliminates the need for dedicated circuitry for BN operation. To avoid input signal distortion of memristor crossbars, HRAT dynamically adjusts the signal magnitude during training using a learned scale factor. These scale factors can be incorporated into the parameters of linear operation together with fused BN, so no additional signal scaling circuits are required.
- We conduct various experiments on baseline networks (FC-4, LeNet-5, and VGG-16) on datasets (MNIST, CIFAR-10, and CIFAR-100) to demonstrate the performance of HRAT. To evaluate the proposed HRAT methodology, FC-4 and LeNet-5 are firstly trained by HRAT and then deployed in hardware. Hardware simulation results match well with the offline HRAT results, indicating that HRAT can bridge the gap between offline training and hardware deployment. To investigate the effectiveness of HRAT on large-scale networks, we conduct experiments using VGG-16 on the CIFAR datasets. Experimental results demonstrate that HRAT can lead to state-of-the-art MNNs without performing prohibitively expensive and time-consuming on-chip retraining, enabling low-cost high-performance MNNs for large-scale commercialization of neuromorphic systems.

## 2 RELATED WORK

**Variation-aware MNN training.** Prior work on MNN offline training has predominately focused on variation-aware training. Liu et al. (2015) address memristor conductance variability (*i.e.*, weight uncertainties) by introducing an additional term called "penalty of variation" into the training constraints. Then, an upper bound for the variation penalty is estimated and used during training. This method improves the tolerance of trained weights to memristor variability by applying tighter training constraints. Zhu et al. (2020) propose statistical training to deal with MNN weight uncertainties. Trained weights are modeled as linear functions of random variables, representing memristor variability. Subsequent network computations are modified to propagate the effects of weight uncertainties. The effectiveness of statistical training has only been validated on small MNNs. Gao et al. (2021) propose a variation-aware MNN training framework, which develops an analytical model for weight uncertainties and uses it as a constraint during training. Yang et al. (2021) propose stochastic-noise-aware training to inject stochastic noise during training. Stochastic noise includes memristor programming noise (*i.e.*, memristor conductance variability), thermal noise, shot noise, and random telegraph noise. Mao et al. (2022) propose defect-aware training to account for the effects of memristor conductance variability, relaxation, and failure during training. Büchel et al. (2022) address the issue of memristor conductance variability via adversarial regularization. In order to enhance the MNN robustness to memristor programming noise, weight space is attacked by adding Gaussian noise Murray & Edwards (1994) to parameters during training. Wan et al. (2022) propose noise-resilient training to inject Gaussian noise into MNN weights during the forward pass. In summary, the aforementioned MNN training methods mainly improve noise immunity to memristor conductance (*i.e.*, weight) uncertainties and address weight mismatch between offline training and on-chip deployment. However, other non-negligible MNN hardware restrictions, such as the non-destructive threshold voltage of memristors, have not been incorporated into training.

**BN fusing strategies for quantization-aware training.** Several BN fusing strategies have been proposed for quantization-aware training (Figure 5 in Appendix for details). By incorporating batch normalization into linear operation (*e.g.*, fully connected layer or convolutional layer), these strategies convert $\boldsymbol{W}$ (*i.e.*, weight of linear operation) into $\boldsymbol{W} \, \boldsymbol{\gamma} / \sqrt{\boldsymbol{\sigma}^2 + \epsilon}$ (*i.e.*, fused weight). Here $\boldsymbol{\gamma}$ is a learnable scale factor in BN, $\boldsymbol{\sigma}^2$ is output variance of linear operation, and $\epsilon$ is a small constant added to prevent the divide by zero error. Wan et al. (2022) implement BN fusion in a straightforward manner Ioffe & Szegedy (2015). Jacob et al. (2018) introduce fused linear operation at the current batch scale. During training, the statistics (*i.e.*, mean and variance) of the current batch are extracted before performing the fused linear operation. Then, these statistics are used to update the Moving Average (MA) statistics and generate fused weight and bias. In this way, a fused linear operation is performed using the obtained fused weight and fused bias. Krishnamoorthi (2018) introduce fused linear operation at the MA scale with a correction for the current batch scale. During training, the output of fused linear operation is corrected from the MA scale to the current batch scale by multiplying $\sqrt{\boldsymbol{\sigma}_{\mathcal{B}}^2 + \epsilon} \Big/ \sqrt{\boldsymbol{\sigma}^2 + \epsilon}$, where $\boldsymbol{\sigma}_{\mathcal{B}}^2$ is the output variance of linear operation extracted from current batch. Thus, an additional linear operation is required to compute the current batch statistics, which is used to update the MA statistics and correct the fused scale of linear operation. PyTorch (2022) introduces fused linear operation at the MA scale with a plain BN and a correction to the unfused scale. In order to apply a plain BN, the scale of fused linear operation is corrected to the the unfused scale by multiplying $\sqrt{\boldsymbol{\sigma}^2 + \epsilon} \Big/ \boldsymbol{\gamma}$. Since the strategies Jacob et al. (2018); Krishnamoorthi (2018) require two linear operations (*i.e.*, fused linear and linear), they are much more computationally expensive than PyTorch (2022) whose strategy involves one linear operation.

While these existing BN fusing strategies help align the distortion caused by weight quantization, the hardware restrictions of MNN systems lead to more aspects of distortion during BN fusion. For example, since fused bias and outputs of fused linear operation are represented as voltages in MNNs, effective BN fusion for MNNs needs to ensure that they conform to the limited output swing of operational amplifiers. Otherwise, voltage saturation and clamping of fused bias or outputs of fused linear operation will cause distortion and inference degradation. Therefore, a BN fusing strategy should align the distortion caused by hardware restrictions between training and hardware inference.

## 3 MNN HARDWARE DEPLOYMENT AND RESTRICTIONS

An MNN consists of multiple interconnected network layers. The hardware schematic for one layer is plotted in Figure 1(a), where a memristor crossbar implements a layer of synapses, and each offline-trained weight is achieved by the difference in memristor conductance between two differential memristors Prezioso et al. (2015); Li et al. (2018); Yao et al. (2020); Wan et al. (2022), allowing the realization of positive and negative weight. Each offline-trained bias is downloaded into a register circuit, which controls a digital-to-analog converter (DAC) for providing a bias voltage to a neuron summation circuit. The neuron summation circuit and activation circuit work together to generate an output voltage Krestinskaya et al. (2019). Figure 1(b) shows the measured current-voltage sweep curve of memristors Yan et al. (2019). Memristors have two distinct modes of operation: a safe operation mode, in which memristor conductance remains unchanged from its previously programmed value, and a conductance programming mode, in which a series of programming pulse voltages are applied to a memristor until its conductance approaches its expected value. Since MNNs should run in the safe operation mode during inference, the input voltage across memristors cannot exceed the upper voltage limit of its safe operation region, such as 0.2V in Yan et al. (2019). This upper voltage limit can be viewed as the non-destructive threshold voltage Jo et al. (2010), $V_{TH}$, below which the previously programmed memristor conductance (*i.e.*, offline-trained weight) does not change during inference. The non-destructive threshold voltage varies with the material, fabrication process, and physical structure of memristors. To ensure proper inference, the input voltage clamp circuit in Figure 1(a) should restrict voltages across the memristor crossbar with $[-V_{TH}, V_{TH}]$.

Figure 1(c) shows the schematic of a neuron summation circuit, which consists of transimpedance amplifiers and fully differential amplifier to convert and scale the current difference (*i.e.*, $I^+ - I^-$) to voltage and then add the DAC generated bias voltage. To avoid the inconvenience caused by dual power supply, the neuron summation circuit is designed to operate with a single positive power supply and set the signal ground level of the neuron summation circuit to be at the half of the supply

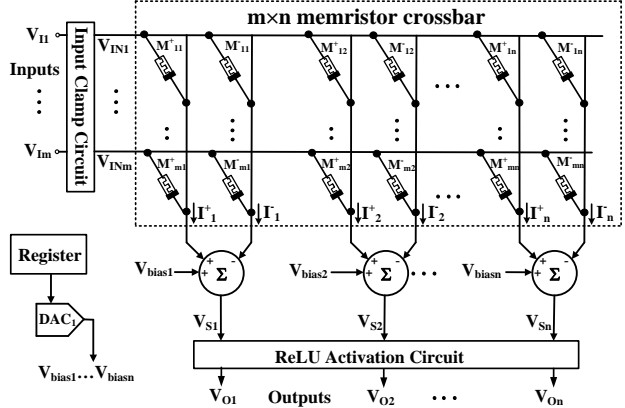
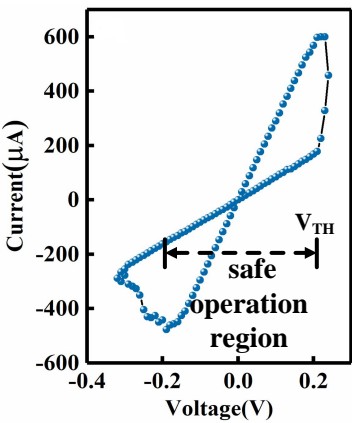

(a) Hardware schematic of one MNN layer for forward pass. A crossbar implements a layer of synapses. Each offline-trained weight is achieved by two differential memristors.

(b) Measured current and voltage sweep curve of memristors Yan et al. (2019)

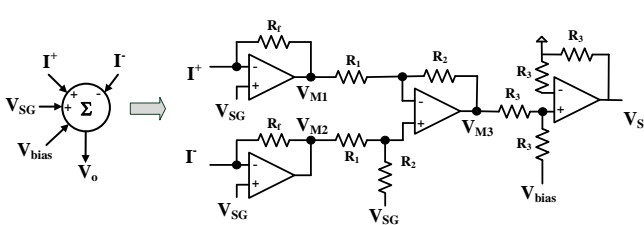
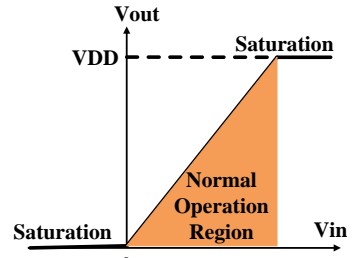

(c) A neuron summation circuit. It uses resistors and operational amplifiers to scale the current difference of differential memristors, and then adds a bias voltage from a digital-to-analog converter (DAC).

(d) Output voltage swing of amplifiers and DACs is limited between 0V and VDD (the supply voltage).

Figure 1: MNN hardware overview and deployment restrictions.

voltage (*i.e.*, $V_{SG}$= VDD$/2$ in Figure 1(c)). In this way, all node voltages are between 0V and VDD (the supply voltage), and the voltages ($V_{IN1}...V_{INm}$) in Figure 1(a) will be clamped into $[V_{SG} - V_{TH}, V_{SG} + V_{TH}]$. In addition to the memristor threshold voltage restriction described above, MNN devices, circuits, and systems suffer from the following restrictions and non-idealities:

**Weight quantization noise.** Offline-trained weight is implemented by memristor conductance in MNN. Memristor is typically programmed to several conductance states, resulting in limited weight bitwidth (*i.e.*, 5 bits Yao et al. (2020) and 4 bits Wan et al. (2022)) and weight quantization noise.

**Weight uncertainty noise.** Due to physical mechanisms (*e.g.*, device relaxation Mao et al. (2022); Wan et al. (2022)) and a limited number of programming pulse cycles (*e.g.*, 500 in Yao et al. (2020)), memristor conductance manifests significant variations, leading to considerable mismatch from offline-trained weight Büchel et al. (2022).

**Limited output swing of operational amplifiers and DACs.** The outputs of the amplifier or DAC can only swing within the power supply range (assuming rail-to-rail circuit topologies are used). The circuit outputs will be clamped at the ground or VDD level when the intended signal values exceed the output swing range as shown in Figure 1(d). Such hardware restriction should be considered during training.

**Bias quantization noise of finite-resolution DACs.** DAC circuits are widely used for bias voltage generation due to their high precision and great flexibility. The output resolution of a DAC (usually specified in bits) represents the smallest output increment that can be produced. When bias voltages are generated by finite-resolution DACs, bias quantization noise potentially degrades the inference performance.

# 4    HARDWARE-RESTRICTION-AWARE TRAINING (HRAT)

In this section, we present Hardware-Restriction-Aware Training (HRAT), which takes into account various non-negligible restrictions of memristor devices, circuits, and systems. Figure 2(a) and 2(b) illustrate the HRAT process for a layer without BN and with the proposed BN fusing strategy, respectively. In HRAT, weight parameters are quantized to mimic the process of using a limited number of pulse cycles to program memristor conductance; bias parameters are quantized to mimic bias quantization noise caused by the finite-resolution of DACs; process variation of memristor devices is mimicked by adding weight uncertainty noise, and the limited output swing of operational amplifiers and DACs is mimicked by a clamp function. Furthermore, a trainable scale factor $s$ is added to each layer. In this way, the output signal magnitude of each network layer can be adjusted to a proper range. Key aspects of HRAT will be described in detail in the following subsections.

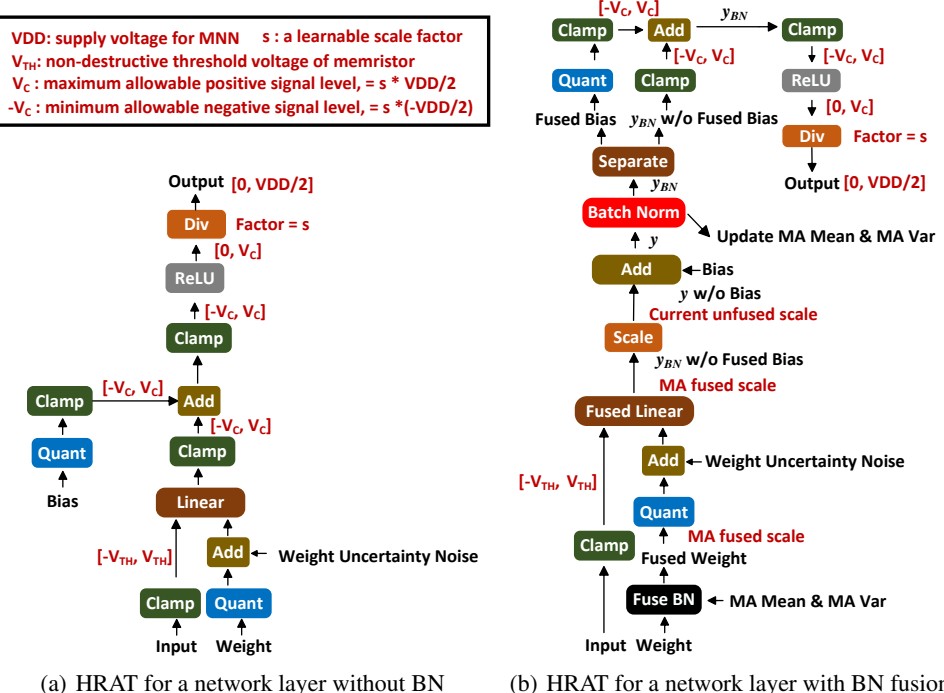

(a) HRAT for a network layer without BN          (b) HRAT for a network layer with BN fusion

Figure 2: Hardware-restriction-aware training (HRAT) for MNNs.

## 4.1    PARAMETER-NOISE-AWARE TRAINING

Instead of training MNNs at full precision, their offline-trained weights are typically quantized to finite discrete states (*i.e.*, 5 bits Yao et al. (2020) and 4 bits Mao et al. (2022); Wan et al. (2022)). As described in Appendix A.1, we uniformly quantize weight parameters and symmetrically clamp the states to a finite number. To model the weight uncertainty and improve the noise robustness of MNNs, we follow Murray & Edwards (1994); Wan et al. (2022) and inject Gaussian noise to weight parameters during forward pass. Thus, weight quantization noise and weight uncertainty noise are added to weight parameters during training. Similarly, due to the limited resolution of DACs, DAC outputs are linearly quantized. Quantization noise is injected into the bias parameter during training. To solve the non-differentiation issue of quantization function, we use the Straight-Through Estimator (STE) trick to relax the quantization function of weight and bias parameters. The perception of weight and bias noise during training is called parameter-noise-aware training, through which the resulting neural networks are insensitive to MNN deployment restrictions of weight quantization, weight uncertainty, and bias quantization noise. While other methods may perform better in addressing parameter noise (*e.g.*, adversarial training Büchel et al. (2022)), simply injecting parameter noise is more computation efficient and sufficient for maintaining MNN accuracy.

### 4.2 NON-DESTRUCTIVE THRESHOLD VOLTAGE OF MEMRISTORS

During inference, a memristor should operate at or below its non-destructive threshold voltage to avoid conductance drift. As the threshold voltage is usually low (*e.g.*, 0.1V Yan et al. (2018) , 0.2V Yan et al. (2019), 0.2V Yao et al. (2020)), the magnitude of most input signals to memristor crossbars will be chopped by the input clamp circuit in Figure 1(a)), causing significant signal distortion. Therefore, HRAT clamps the signal magnitude across memristor crossbars to $[-V_{TH}, V_{TH}]$ during training to correspond to the input clamp circuit in hardware MNNs.

### 4.3 SIGNAL MAGNITUDE SCALING

Note that all output voltages of operational amplifers and DACs in Figure 1 are restricted between 0 and VDD. By setting the signal ground level $V_{SG}$=VDD/2 in the neuron summation circuit of Figure 1(c), the equivalent signal swing range is $[-$ VDD$/2$ , VDD$/2]$. Then, by introducing a learnable scale factor $s$ for each network layer in the proposed HRAT, the equivalent signal swing range at these hardware nodes is clamped to $[-s \cdot$ VDD$/2$ , $s \cdot$ VDD$/2]$. Overall, HRAT divides output signals of a network layer by a scale factor $s$ and attempts to linearly compresses them within the output swing of operational amplifiers and DACs. During inference, these scale factors can be incorporated into the parameters of linear operation together with fused BN, so no additional signal scaling circuits are required.

### 4.4 BN FUSING STRATEGY

Training an MNN with or without BN layers makes a big difference. The training process for a network layer without BN operation is illustrated in Figure 2(a), which takes into account the above hardware restrictions. Regarding training a network layer with BN operation, we propose a new BN fusing strategy to align the restriction-induced distortion between offline training and hardware inference. Referring to PyTorch (2022), this new BN fusing strategy performs one linear operation. As illustrated in Figure 2(b), the fused weight is first obtained using the MA statistics, and then the weight quantization noise and weight uncertainty noise are injected into the fused weight. Next, after performing the fused linear operation, the output is corrected to the unfused scale by multiplying $\sqrt{\boldsymbol{\sigma^2 + \epsilon}}/\boldsymbol{\gamma}$. Because the corrected output is on the unfused scale, normal BN can be performed to obtain the result $y_{BN}$ and update the MA statistics. Since the calculation of $y_{BN}$ does not consider hardware restrictions, $y_{BN}$ is separated into two terms: fused bias $\boldsymbol{b}_{fused}$ and BN result without fused bias '$y_{BN} - \boldsymbol{b}_{fused}$'. Next, bias quantization noise is added into the $\boldsymbol{b}_{fused}$. Then, quantized $\boldsymbol{b}_{fused}$, '$y_{BN} - \boldsymbol{b}_{fused}$' and their sum are clamped to the range of $[-s \cdot$ VDD$/2$ , $s \cdot$ VDD$/2]$, respectively. After passing a ReLU activation function and dividing a scale factor $s$, the final output of HRAT for a network layer is limited to $[0,$ VDD$/2]$.

## 5 BASELINE NETWORK ARCHITECTURES AND EXPERIMENTAL SETUP

We choose a four-layer fully connected NN (FC-4), LeNet-5 LeCun et al. (1998), and VGG-16 Simonyan & Zisserman (2015), as our baseline architectures. We design the miniature model FC-4 for rapid verification of the proposed HRAT algorithm. FC-4 consists of three hidden layers and one classification layer. The three hidden layers have 512, 128, and 32 nodes respectively. LeNet-5 and VGG-16 are implemented from the original papers, only slightly different to accommodate different datasets. For experiments on the CIFAR datasets, the number of features in the hidden layers of LeNet-5 is increased by a factor of 5. For fast convergence and better performance, we also implement batch normalization (BN) layers in each baseline model. Features are normalized via BN except for the fully connected layers in VGG-16. The default supply voltage of MNNs is 3V. Memristor behaviors (*e.g.*, the non-destructive threshold voltage of 0.2V and conductance tuning range of [2$\mu$S, 20$\mu$S]) are obtained from the experimental results Yao et al. (2020). Offline weights are initialized and limited to the range of [-1,1]. After HRAT, offline-trained weights are transformed to the memristor conductance values for hardware deployment. Weight noise follows a Gaussian distribution and is generated according to the weight range, for example, std=0.1 means that the standard deviation of weight noise is equal to 10% of the entire weight range. We use 40 test runs to statistically measure the inference performance of these baseline architectures.

# 6 EXPERIMENTAL RESULTS AND DISCUSSION

## 6.1 FC-4 AND LENET-5 ON MNIST

Figure 3(a) plots the mean inference accuracy of FC-4 on MNIST for several combinations of weight noise and weight bitwidth. An inference accuracy of 98.60% is obtained using the software benchmark models (*i.e.*, floating-point or 8-bit quantized weights without MNN hardware restrictions). Mean accuracies of 96.05% and 97.82% are achieved for HRAT without and with signal strength scaling, respectively. The 1.77% accuracy difference reflects the importance of performing signal magnitude scaling in HRAT. If on-chip retraining is performed after HRAT, the mean accuracy rises from 97.82% to 98.40%, which is very close to the software benchmark result (*i.e.*, 98.60%). Although on-chip retraining has been demonstrated in small-scale MNNs Li et al. (2018); Wang et al. (2019); Yao et al. (2020) to recover the accuracy drop caused by hardware non-idealities, such approaches require complex analog backpropagation learning circuitry Krestinskaya et al. (2018a;b), making them unsuitable for cost-effective hardware implementation. Figure 3(b) plots the standard deviation (std) of inference accuracy for FC-4 on MNIST. HRAT results in an average variance of inference accuracy of 0.39, which drops to 0.07 after performing on-chip retraining. Figure 3(c) plots the mean inference accuracy of LeNet-5 on MNIST for several combinations of power supply voltage VDD and memristor threshold voltage $V_{TH}$, assuming zero weight noise and 8-bit quantized weights. Thanks to signal magnitude scaling, HRAT is insensitive to the choice of power supply voltage and memristor threshold voltage, and hence achieves a mean accuracy of 98.84%. Figure 3(d) shows very close inference accuracy of FC-4 and LeNet-5 models on MNIST with HRAT.

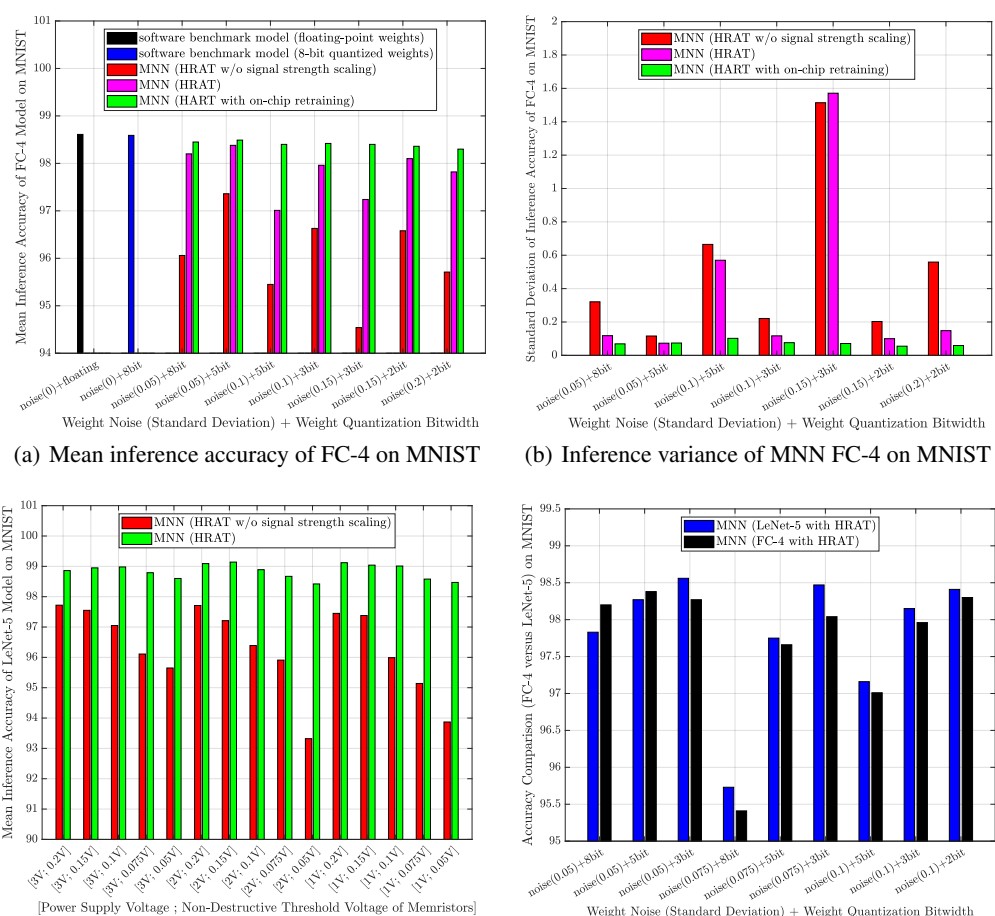

(a) Mean inference accuracy of FC-4 on MNIST

(b) Inference variance of MNN FC-4 on MNIST

(c) Mean inference accuracy of MNN LeNet-5 on MNIST for several combinations of VDD and $V_{TH}$

(d) Approximate inference accuracy of FC-4 and LeNet-5 models on MNIST with HRAT

Figure 3: Inference performance of MNN FC-4 and LeNet-5 models on MNIST.

To validate the proposed HRAT, offline trained FC-4 and LeNet-5 models are implemented in hardware circuits and simulated using the Cadence Spectre tool. Fully-connected layers are implemented according to Figure 1(a) (Figure 6 in Appendix for details). For convolutional layers, each filter kernel is implemented by a sub-circuit similar to a fully-connected layer. The filter sub-circuit is instantiated multiple times, so all neuron convolutions of a layer are computed simultaneously. ReLU activation and max pooling are also implemented by dedicated analog circuits. BN fused weights and biases, along with scale parameters obtained from HRAT are transformed to memristor conductance values, DAC outputs, and amplifier gains in these analog circuits. To reduce hardware simulation time, we use a macro model of operational amplifiers to capture realistic behaviors, such as limited output swings, finite voltage gain, limited bandwidth, etc. Then, output voltages of each neuron obtained from hardware circuit simulations are compared with their corresponding offline HRAT results. We find hardware simulation matches with offline HRAT. (Appendix A.4 for details)

## 6.2 VGG-16 on CIFAR datasets

Figure 4(a) plots the mean inference accuracy of VGG-16 on CIFAR-100 for several combinations of weight noise and weight bitwidth. The software benchmark models (*i.e.*, floating-point weights or 8-bit quantized weights) achieve inference accuracies of 68.59% and 66.27%, respectively. Note that software benchmark models are not affected by hardware non-idealities. At a weight noise level of std = 0.025, HRAT with 6-bit quantized weight leads to the highest accuracy of 62.85%. When the weight noise level is std = 0.05, HRAT with 4-bit quantized weight leads to the highest accuracy of 57.94%, which is only 8.33% lower than the software benchmark model (*i.e.*, 66.27%). Note std=0.05 means that the standard deviation of Gaussian weight noise is equal to 5% of the entire weight range, this weight noise level is significant. These results demonstrate that HRAT is robust to weight noise disturbance. Figure 4(a) also reveals that, for a given weight noise level, there is an optimal weight bitwidth to balance the trade-off between noise immunity and expressiveness of MNN. Lower bitwidth means higher quantization noise is injected into weight during training, thus exhibiting stronger noise immunity during inference. However, lower bitwidth is not always better, because the expressive power of the model is limited Yoon et al. (2022). This explains why the optimal weight bitwidth in Figure 4(a) tends to be lower at stronger noise levels. Furthermore, Figure 4(a) shows that if the signal magnitude scaling is not performed in HRAT, training cannot converge at all. Compared to Figure 3(a), where a 1.77% drop in accuracy for FC-4 trained by HRAT without signal magnitude scaling, we can see that signal magnitude scaling is more critical and indispensable in large-scale MNNs. As shown in Figure 4(a), if VGG-16 is retrained on-chip after HRAT, the best accuracy rises to 65.93% and 65.88% for the weight noise levels of std=0.025 and std=0.05, respectively. Both accuracies are close to the software benchmark result (*i.e.*, 66.27%). Figure 4(a) also shows that the optimal weight bitwidth for on-chip retraining is 8. This is because on-chip retraining is done individually for each deployed MNN, instead of statistically like HRAT. Therefore, for each VGG-16 model under on-chip retraining, weight uncertainty caused by memristor variations becomes deterministic rather than stochastic. As a result, when training a model with deterministic weight uncertainty, the higher the weight bitwidth, the better the retraining result.

Figure 4(b) plots the variance of inference accuracy for VGG-16 on CIFAR-100. As lower bitwidth is more robust to noise disturbance, HRAT with lower bitwidth leads to less variance in inference accuracy. For HRAT with on-chip retraining, since retraining is performed individually with deterministic weight uncertainty, higher bitwidth results in less variance in inference accuracy.

The effect of DAC resolution on HRAT is simulated and depicted in Figure 4(c) and 4(d), where the performance of software benchmark models is also plotted for comparison. Both figures demonstrate that the use of finite-resolution of DACs has a big impact on inference. If an ideal DAC (*i.e.*, infinitely small resolution) is used to generate bias, an increase in weight noise reduces the inference accuracy. For a fixed weight noise level, the reduction in DAC resolution significantly deteriorates inference. Figure 4(e) shows that very close results can be obtained using 14-bit or ideal DACs (*i.e.*, infinitely small resolution) for bias generation. To maintain high inference accuracy, DAC resolution for VGG-16 on CIFAR-10 and CIFAR-100 needs to be no lower than 9 or 12 bits, respectively.

To investigate the effect of using learnable scale factors on HRAT, Figure 4(f) plots the inference accuracy curves of VGG-16 over 500 epochs on CIFAR-100. Compared with simulation curves using fixed scale factors in HRAT, the learnable scale factors help VGG-16 to converge with 2.42% and 4.71% higher accuracy for ideal DACs and 11-bit DACs, respectively.

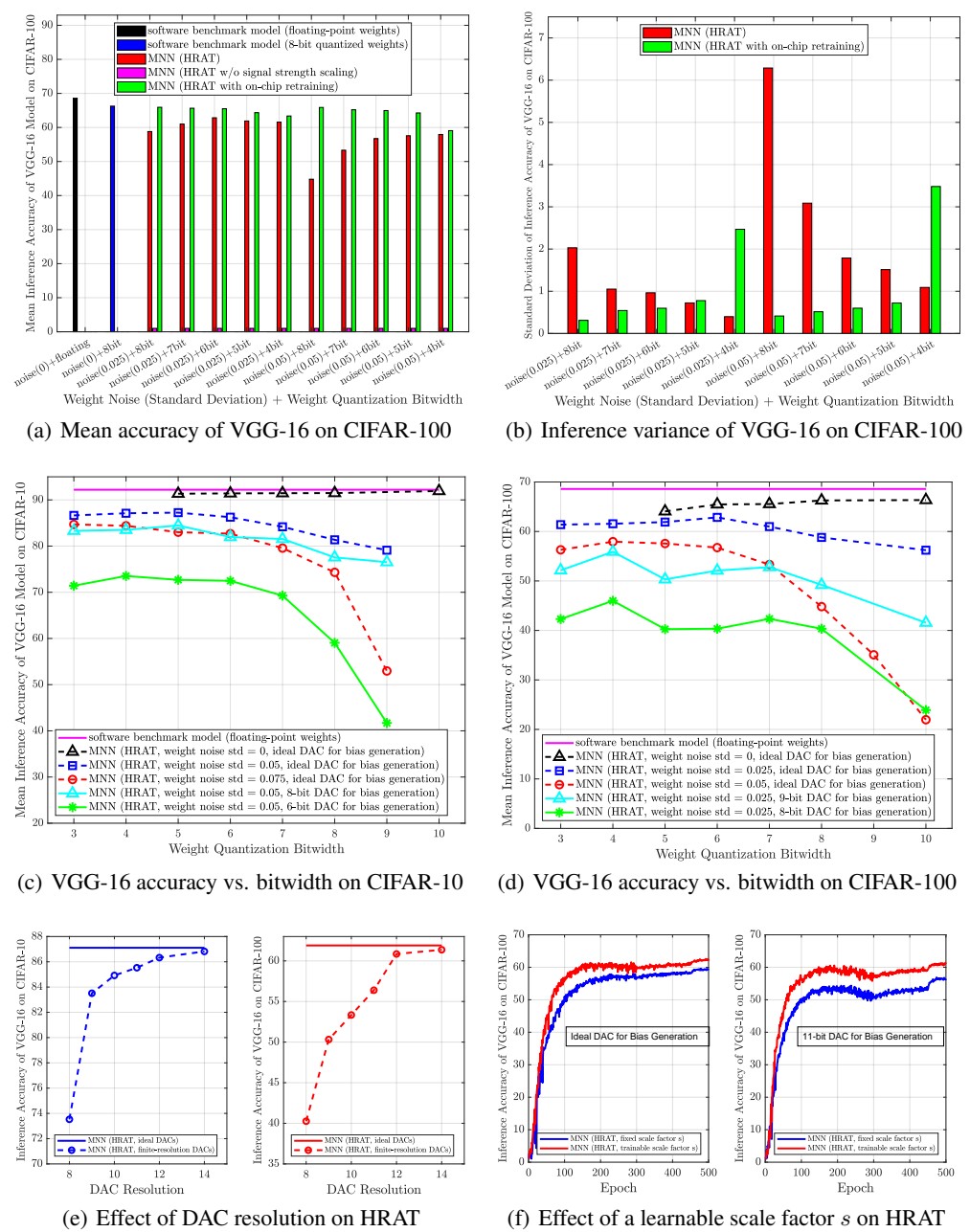

(a) Mean accuracy of VGG-16 on CIFAR-100

(b) Inference variance of VGG-16 on CIFAR-100

(c) VGG-16 accuracy vs. bitwidth on CIFAR-10

(d) VGG-16 accuracy vs. bitwidth on CIFAR-100

(e) Effect of DAC resolution on HRAT

(f) Effect of a learnable scale factor $s$ on HRAT

Figure 4: Inference performance of MNN VGG-16 models on CIFAR datasets.

## 7 CONCLUSION

We propose hardware-restriction-aware training (HRAT) for memristor neural networks (MNNs) to consider non-negligible hardware restrictions. HRAT integrates various hardware restrictions, adopts a new BN fusing strategy, and dynamically adjusts signal magnitude to avoid distortion. The simulation results of MNN hardware implementation match well with HRAT results, validating that HRAT successfully mimics the realistic behavior and hardware restrictions of MNNs during offline training. Experimental results also demonstrate that HRAT can lead to state-of-the-art MNNs without performing prohibitively expensive and time-consuming on-chip retraining, enabling low-cost high-performance MNNs for large-scale commercialization of neuromorphic computing systems.

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

## A  APPENDIX

### A.1  WEIGHT QUANTIZATION

To simplify the conductance programming of memristor crossbars, weights are quantized into a finite number of states. Low-precision weight makes the model insensitive to deployment noise. Lower bitwidth means higher quantization noise is injected to weight during training. Because weight quantization is deterministic, only uncertainty noise causes weight distortion after on-chip deployment. Lower bitwidth does not always lead to better results, since it limits the representative power of neural networks.

In this work, we uniformly quantize weights using the following quantization function:

$$\mathcal{Q}(\boldsymbol{W}; bw) = \mathcal{C}_{bw}(\lfloor \frac{\boldsymbol{W}}{\Delta} \rceil \Delta) \tag{1}$$

where $\Delta$ is a trainable parameter of the quantization function to denote the quantization step size, $\lfloor \cdot \rceil$ denotes the rounding operation, $bw$ is a given bitwidth, and $\mathcal{C}_{bw}(\cdot)$ clamps quantized values into the range of $[-(2^{bw-1} - 1) \cdot \Delta, (2^{bw-1} - 1) \cdot \Delta]$. Due to the non-differentiation of rounding operation, weight and quantization function cannot be trained through gradient-based optimizers. To address this issue, Eq. (1) is converted into a differentiable form:

$$\mathcal{Q}(\boldsymbol{W}; bw) = \mathcal{C}_{bw}((\frac{\boldsymbol{W}}{\Delta} + \delta_{rounding}) \cdot \Delta) \tag{2}$$

where the rounding noise $\delta_{rounding} = \lfloor \frac{W}{\Delta} \rceil - \frac{W}{\Delta}$. We assume that $\delta_{rounding}$ is an independent noise, so the gradient of the non-differential $\delta_{rounding}$ is not calculated during back-propagation. This method is also known as the Straight-Through Estimator (STE) trick. Thus, the gradient of $W$ and $\Delta$ is obtained through the back-propagation algorithm. Therefore, we can easily train memristor neural networks with gradient-based optimizers.

### A.2 EXISTING BN FUSING STRATEGIES

A linear layer with BN is expressed as:

$$y = Wx + b \tag{3}$$

and

$$y_{BN} = \gamma \frac{y - \mu_{\mathcal{B}}}{\sqrt{\sigma_{\mathcal{B}}^2 + \epsilon}} + \beta \tag{4}$$

Where $x$ is the input, $W$ is the weight parameter, $b$ is the bias term, $\gamma$ is the learnable scale factor of BN, $\beta$ is the bias term of BN, $\mu_{\mathcal{B}}$ is mean output of the current batch, and $\sigma_{\mathcal{B}}^2$ is the variance of the current batch. During inference, MA statistics $\mu$ and $\sigma^2$ are used for normalization. For full precision software models, BN is fused into the linear operation via combining (3) and (4). Then, the inference can be calculated with fused parameters (fused weight and fused bias) as:

$$y_{BN} = \frac{\gamma}{\sqrt{\sigma^2 + \epsilon}} Wx + \beta + \frac{\gamma}{\sqrt{\sigma^2 + \epsilon}}(b - \mu) \tag{5}$$

where the fused weight $W_{fused} = \frac{\gamma}{\sqrt{\sigma^2 + \epsilon}} W$, and the fused bias $b_{fused} = \beta + \frac{\gamma}{\sqrt{\sigma^2 + \epsilon}}(b - \mu)$.

During training, existing BN fusing strategies are illustrated in Figure 5. Figure 5(a) shows that instead of fusing BN during training, BN is fused during deployment according to Eq. (5). Wan et al. (2022) train their MNN in this way. Figure 5(b) 5(c) 5(d) illustrate the training schemes for quantization-aware training (QAT). They assume that the quantization effect of fused bias can be ignored, so only consider the distortion caused by the fused weight quantization. During inference, only fused linear operation needs to be performed using fused weight and bias obtained from the MA statistics.

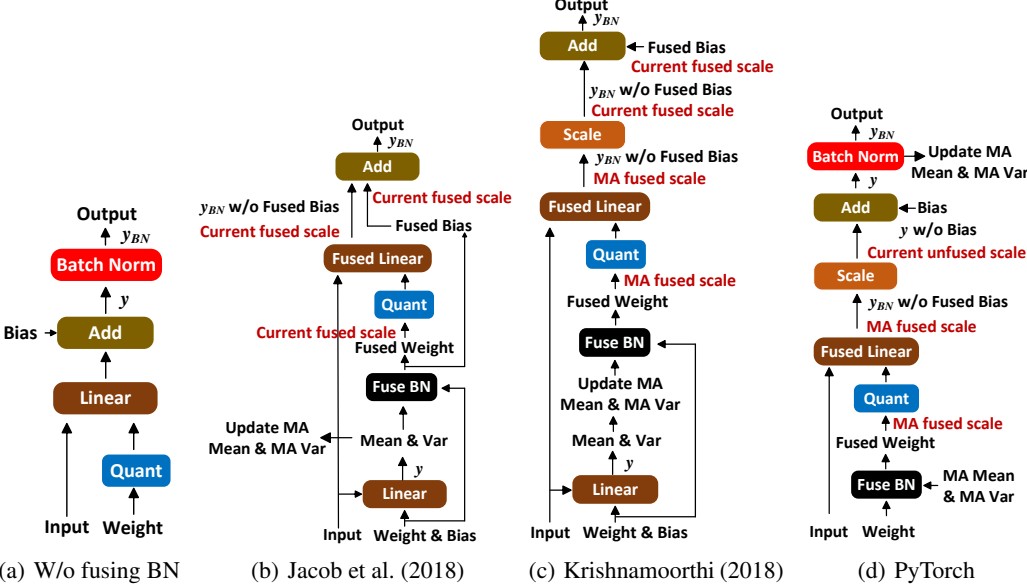

Figure 5: Existing BN fusing strategies in literature.

### A.3 FORMULATIONS OF HRAT

### A.3.1 TRAINING WITHOUT BN

As illustrated in Fig. 2(a), training a fully-connected layer without BN is expressed as:

$$\boldsymbol{y} = \frac{\text{ReLU}(\mathcal{C}_{V_c}(\mathcal{C}_{V_c}((\mathcal{Q}(\boldsymbol{W}; bw) + \boldsymbol{W}_{noise})\mathcal{C}_{V_{TH}}(\boldsymbol{x})) + \mathcal{Q}(\boldsymbol{b})))}{s} \tag{6}$$

where $\mathcal{C}_{V_c}(\cdot)$ clamps input to the scaled signal range of $[-V_c, V_c]$, $V_c = s \cdot \text{VDD}/2$, $\boldsymbol{W}_{noise}$ is generated from a Gaussian distribution to model memristor variations, $s$ is a learnable scale factor. $\mathcal{Q}(\boldsymbol{b})$ quantizes $\boldsymbol{b}$ according to the bitwidth of DAC and the scaled signal range of $[-V_c, V_c]$. In fact, we use uniform noise instead of quantization noise to introduce randomness.

After training, weight is divided by $s$ to obtain scaled weight. Since a memristor usually has a certain conductance range to tune (e.g., $[2\mu\text{S}, 20\mu\text{S}]$ Yao et al. (2020)), scaled weight is transformed to a memristor conductance by multiplying a converting factor $s_c$ as:

$$\boldsymbol{W}_{deploy} = s_c \cdot \frac{\mathcal{Q}(\boldsymbol{W}; bw)}{s} \tag{7}$$

Bias is divided by $s$ to fuse the scale factor.

$$\boldsymbol{b}_{deploy} = \frac{\mathcal{Q}(\boldsymbol{b})}{s} \tag{8}$$

Then, we can program $\boldsymbol{W}_{deploy}$ to memristor crossbars and program $\boldsymbol{b}_{deploy}$ to DACs. Since weight is transformed to the memristor conductance range, an additional signal amplification factor $1/s_c$ should be taken into account. Therefore, the inference is expressed as:

$$\boldsymbol{y} = \text{ReLU}(\mathcal{C}_{\text{VDD}/2}(\mathcal{C}_{\text{VDD}/2}(1/s_c \cdot (\boldsymbol{W}_{deploy} + \boldsymbol{W}_{noise})\mathcal{C}_{V_{TH}}(\boldsymbol{x})) + \boldsymbol{b}_{deploy})) \tag{9}$$

### A.3.2 TRAINING WITH THE PROPOSED BN FUSING STRATEGY

Figure 2(b) shows a fully-connected layer trained with the proposed BN fusing strategy. Deployment noise is applied on fused weight, through introducing a factor $\frac{\gamma}{\sqrt{\sigma^2 + \epsilon}}$, we fuse weight as:

$$\boldsymbol{W}_{fused} = \frac{\gamma}{\sqrt{\sigma^2 + \epsilon}} \cdot \boldsymbol{W} \tag{10}$$

In order to apply a normal BN after fused linear operation, we correct the output to original scale by dividing $\frac{\gamma}{\sqrt{\sigma^2 + \epsilon}}$, and then the bias term is added to the output of original scale. Before applying BN, we have:

$$\boldsymbol{y} = \mathcal{Q}(\boldsymbol{W_{fused}}; bw) + \boldsymbol{W}_{noise})\mathcal{C}_{V_{TH}}(\boldsymbol{x}) + \mathcal{Q}(\boldsymbol{b}) \tag{11}$$

The result $y_{BN}$ is obtained by applying a normal BN according to Eq. (4). The above steps only address fused weight, other hardware restrictions are still not integrated. In order to quantize fused bias and clamp the output range, we separate $y_{BN}$ to fused bias $\boldsymbol{b}_{fused}$ and $\bar{y}_{BN}$.

$$\boldsymbol{b}_{fused} = \boldsymbol{\beta} + \frac{\gamma}{\sqrt{\sigma^2 + \epsilon}}(\boldsymbol{b} - \boldsymbol{\mu}) \tag{12}$$

$$\bar{y}_{BN} = y_{BN} - \boldsymbol{b}_{fused} \tag{13}$$

Then, other hardware restrictions are applied based on the illustration in Figure 2(b). After training, the BN and $s$ are fused into previous linear operation. So we have

$$\boldsymbol{W}_{deploy} = s_c \cdot \frac{\mathcal{Q}(\frac{\gamma}{\sqrt{\sigma^2 + \epsilon}}\boldsymbol{W}; bw)}{s} \tag{14}$$

$$\boldsymbol{b}_{deploy} = \frac{\mathcal{Q}(\boldsymbol{b}_{fused})}{s} \tag{15}$$

Finally, we can perform hardware deployment, similar to the steps in Section A.3.1.

### A.4 CIRCUIT SIMULATION RESULTS

Figure 6 shows the hardware operation of a fully-connected layer. The input signal is clamped to avoid excessive voltage across memristors. Linear operation is executed via a memristor crossbar, where weights are programmed into memristor conductance values. The output of linear operation, along with bias voltages from DACs, are passed to a neuron summation circuit and activation circuit for processing. Table 1 summarizes the difference in results between offline HRAT and hardware circuit simulation for all nodes across all layers of LeNet-5. The second column lists the number of outputs at each layer. We use the MNIST dataset with an input dimension of $28 \times 28 \times 1$ for this experiment. Hence, the output dimension of the first convolutional layer is $24 \times 24 \times 6$. The last three columns summarize the maximum, mean, and standard deviation of the error values for each network layer. It demonstrates that circuit simulation results closely match the offline HRAT results.

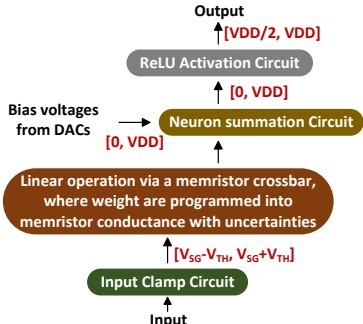

Figure 6: Schematic of a fully-connected layer for memristor neural network.

Table 1: All-layer output comparison of LeNet-5 between offline HRAT and hardware simulation.

| network layer | # of outputs | Maximum Error | Mean error | Error variance std |
|---|---|---|---|---|
| conv. layer 1 | 3456 | $1.40 \times 10^{-3}$ | $1.24 \times 10^{-4}$ | $1.85 \times 10^{-4}$ |
| conv. layer 2 | 1024 | $3.56 \times 10^{-3}$ | $3.97 \times 10^{-4}$ | $5.82 \times 10^{-4}$ |
| fc layer 1 | 120 | $2.29 \times 10^{-3}$ | $3.33 \times 10^{-4}$ | $5.15 \times 10^{-4}$ |
| fc layer 2 | 84 | $4.10 \times 10^{-3}$ | $7.83 \times 10^{-4}$ | $1.07 \times 10^{-3}$ |
| fc layer 3 | 10 | $8.00 \times 10^{-2}$ | $1.59 \times 10^{-2}$ | $2.73 \times 10^{-2}$ |

### A.5 SIMULATION RESULTS OF VGG-16 ON CIFAR DATASETS

Table 2 lists the mean inference accuracy of VGG-16 on CIFAR datasets, when four weight noise levels (std=0, std=0.025, std=0.5, and std=0.075) are present. VGG-16 is either trained by HRAT, or by HRAT followed by on-chip retraining. The optimal weight bitwidth is selected to report the highest inference accuracy for each case. The inference accuracy of HRAT slowly drops as the weight noise level increases. At a strong weight noise level (std=0.05), HRAT achieves a high accuracy of 87.24% for CIFAR-10 dataset. If on-chip retraining is applied after HRAT, the inference accuracy is close to the baseline (*i.e.*, weight noise std=0) results.

Table 2: Performance comparison among various weight noise levels for VGG-16 on CIFAR datasets, assuming very high-resolution DACs are used for bias generation.

| | CIFAR-10 | | | CIFAR-100 | | |
|---|---|---|---|---|---|---|
| Weight Noise (std) | HRAT | HRAT+retrain | Diff | HRAT | HRAT+retrain | Diff |
| 0.0 (baseline) | 91.48 | 91.48 | 0.0 | 66.27 | 66.27 | 0.0 |
| 0.025 | 90.33 | 91.30 | 0.97 | 62.85 | 65.93 | 3.08 |
| 0.05 | 87.24 | 90.72 | 3.48 | 57.94 | 65.88 | 7.94 |
| 0.075 | 84.70 | 90.70 | 6.00 | 55.64 | 66.10 | 10.46 |

