# OpenReview forum: "Hardware-restriction-aware training (HRAT) for memristor neural networks"
_ICLR.cc/2023/Conference — Submitted to ICLR 2023_

### Official Review · Reviewer_dKMU · 2022-10-21

**Confidence:** 2
**Correctness:** 3
**Technical Novelty And Significance:** 2
**Empirical Novelty And Significance:** 2
**Recommendation:** 3

**Clarity, Quality, Novelty And Reproducibility:**

This paper explores a challenging and important topic, which is the design and training of MNNs. However, it is difficult for me to tell apart the preliminaries and the contribution of this paper in Section 3 and 4. In terms of reproducibility, I cannot find the details to reproduce the results reported in the experiments.

**Strength And Weaknesses:**

Pros:

++ This paper targets an interesting topic and could benefits the community of efficient community.

++ The experiments on various models and datasets show the effectiveness.

Cons:

-- The proposed hardware-restriction-aware training seems to adopt the major framework of previous work [1,2,3], such as non-destructive threshold voltage of memristors and parameter-noise-aware training. It is difficult to tell apart the contribution of the authors and the preliminaries.

-- MNNs can achieve better energy efficiency, however, there is no relevant evaluation of the trade-offs between accuracy and efficiency compared with other efficient networks in the experiment.

-- Lack of comparison with other SOTA basslines of MNNs, such as [4,5], in Figure 3 and 4.

-- Although the author highlights the design of BN fusing, it is difficult to see its superiority besides signal strength scaling.


[1]. Yan et al. Density effects of graphene oxide quantum dots on characteristics of zr0. 5hf0. 5o2 film memristors. Applied Physics Letters, 114(16):162906, 2019.

[2]. Jo et al. Nanoscale memristor device as synapse in neuromorphic systems. Nano letters, 10(4):1297– 1301, 2010.

[3]. Wan et al. A compute-in-memory chip based on resistive random-access memory. Nature, 608(7923):504–512, 2022.

[4]. Yao et al. Fully hardware-implemented memristor convolutional neural network. Nature, 577 (7792):641–646, 2020.

[5]. Wang et al. In situ training of feed-forward and recurrent convolutional memristor networks. Nature Machine Intelligence, 1(9):434–442, 2019.

**Summary Of The Paper:**

The authors introduce a batch normalization fusing strategy to align the distortion between offline training and hardware inference for Memristor neural networks (MNNs). Through a learnable scaling factor, the proposed training algorithm can adjust the magnitude of input signal adaptively. The evaluation is conducted on various models and datasets. The comparison shows that the signal strength scaling plays an important role in recovering the accuracy.

**Summary Of The Review:**

As mentioned above, more clarification and evaluation are required to make this paper more convincing.

---

### Official Review · Reviewer_T82t · 2022-10-25

**Confidence:** 4
**Correctness:** 2
**Technical Novelty And Significance:** 2
**Empirical Novelty And Significance:** 1
**Recommendation:** 3

**Clarity, Quality, Novelty And Reproducibility:**

(1) What are the main contributions of your work? And how do you compare/contrast your work with prior work (some listed below)? Especially, what are the trade-offs between using your approach vs. relying on in-Situ training methods?

[a] [Stable and compact design of Memristive GoogLeNet Neural Network](https://www.sciencedirect.com/science/article/abs/pii/S0925231221002290)

[b] [Memristor-Based Multilayer Neural Networks With Online Gradient Descent Training](https://ieeexplore.ieee.org/abstract/document/7010034)

(2) Section A.4 presents circuit simulation results at layer-granularity. Are there any additive behavior for each layer error values? That is, do you pass on the error from one layer to another layer? How does the final accuracy changes when considering the overall circuit non-idealities?

(3) How does your approach addresses the PVT-variations for the target hardware? Do you suggest multiple training per each designed hardware? How do you plan to incorporate these non-idealities into your training workflow?

(4) In a similar vein, as the memristor non-idealities could change as the circuit ages, how does your approach mitigate the associated noise in these scenarios? Do you think a hybrid online/offline approach is more practical?

(5) How does your network scales for larger NNs? In one of the suggested references, the authors explored GoogleNet as their case study. Is there any limitations/challenges to use your method for these semi-large NNs?

(6) Do you use the exact same circuit non-idealities during training and hardware simulations? How sensitive your approach is to small variations in the induced noise?

**Strength And Weaknesses:**

**Strength**

- The paper nicely reviews some of the prior work on memristor training and assembles the existing methods into an end-to-end framework for training.

- Results across few small-sized NNs show that the suggested training approach could increase the final model performance.


**Weaknesses**

- While the paper advocates for an offline-training method, the trade-offs between an online and offline (or possibly hybrid approach) is not neither well-studied, nor justified.

- While there are few work that have explored and analyzed memristor training for semi-large models, the paper lacks showing the generality and scalability of their solution for these models.

- Using the same noise distribution during training and deployment is not practical. This is because not only each circuit has its own non-idealities but also these non-idealities change as circuit ages.

**Summary Of The Paper:**

The paper advocates for an offline training mechanism for memristor devices and proposes a customized training approach (employing noise-aware training and batch normalization) for such devices. The results across a set of small-sized NNs show that using BN and this hardware-restriction-aware training improves the final model accuracy.

**Summary Of The Review:**

The paper shows the feasibility of employing noise-aware/hardware-limitation-aware training for memristor circuits and show limited set of results for few small-sized NNs. While I agree that exploring non-conventional devices for future generation of NN accelerators is crucial, but I am not convinced that the paper brings any additional insights to the existing literature in terms of training under such hardware constraints.

---

### Official Review · Reviewer_SYfm · 2022-10-25

**Confidence:** 3
**Correctness:** 3
**Technical Novelty And Significance:** 3
**Empirical Novelty And Significance:** 3
**Recommendation:** 5

**Clarity, Quality, Novelty And Reproducibility:**

The authors present useful Figures and Equations (in Appendix) that help understand the ideas well. Adding a sample inference only dataflow might help improve the readability of the requirements added due to the Fused BN step going to MA scaled and Current scaled domains.


**Strength And Weaknesses:**

The authors have done a good job of explaining the context of the problem (Section 3) and the prior art (Section 2). Additionally, Section 6 presents a useful set of experimental results that help understand the proposed methodology and its limitations (on-chip training results included). Especially, Section 6.2 presents useful explanation of the analysis on large network which helps explain the inference accuracy trends reported in Figure 3 and 4.

In Section 6.1 the authors mention that HRAT simulated results match the voltages measured in memristor hardware simulations to a good accuracy. While Table 1 in Appendix A.4 captures this correlation, it would be helpful if the authors added relative accuracy metrics instead of absolute. Depending on the layer in question and features involved the same absolute magnitude of error can have a large range of impact, please add % error stats instead. Also, can the authors explain why the results observed for FC-4 and LeNet-5 in terms of HRAT to HW correlation will hold for larger networks?

Figure 3 highlights the superior training accuracy of HRAT and its variants, it would be useful to have a comparison against other training approaches for these data points as it would reveal the additional accuracy improvement enabled by HRAT.

Section 6.2, the authors mention that std=0.05 is a significant noise range for which HRAT suffers an ~8% reduction on accuracy compared to software trained model. This assertion needs two qualifiers, can the authors add more details justifying the claim that 5% is a typical/large value for the observed noise and secondly, 8% loss in accuracy can be significant depending on the actual application involved. Specially considering that the alternative of on-chip retraining though costly can give very high accuracy with low variance. It would again benefit the discussion to add other approaches for MNN training that don’t involve on-chip training and highlighting HRAT’s accuracy improvement over them.

Section 6.2 explains that limited DAC resolution plays an important role in dictating the accuracy loss in HRAT. Trainable s-factor scales down this noise significant and allows selecting the sweet spot of weight quantization for given noise levels for best accuracy. Basic question for the authors, does the limited DAC resolution scenario have no-impact in the case of on-chip training? If it does and HRAT still achieves comparable accuracy it might be helpful to add that in Figure 4.


**Summary Of The Paper:**

The paper presents a new training methodology, HRAT, for neural networks to be deployed on Memristor (RRAM) based hardware. The presented methodology accounts for non-idealities such as Quantization error for weights and limited output swing of operational amplifier circuits. Additionally, the methodology also covers training networks with fused batch normalization (fuse BN step with previous linear operation). The authors account for additional errors potentially introduced in different steps involved in their fused BN approach. Lastly, the authors present results showing models trained with HRAT demonstrate high accuracy despite hardware limitations for small scale networks. They also highlight the scalability of HRAT to larger networks showing good inference accuracy despite large quantization noise ratios. They also highlight the effectiveness of the scale factor parameter integral to HRAT methodology that enables high inference accuracy.

**Summary Of The Review:**

The authors have done a good job explaining the key ideas and running experiments to demonstrate their effectiveness. Adding a few more baseline points to the comparisons might help the authors better justify the impact of their proposed technique. The trainable s-factor parameter is very useful as it allows authors to selectively tune out the noisiest blocks in the hardware implementation of their design. The key hardware knowledge used by the authors to model all sources of error in MNNs is another useful contribution of the paper.

---

### Official Review · Reviewer_Xqdg · 2022-10-30

**Confidence:** 5
**Correctness:** 2
**Technical Novelty And Significance:** 2
**Empirical Novelty And Significance:** 2
**Recommendation:** 3

**Clarity, Quality, Novelty And Reproducibility:**

The paper is well written.

However, the work is far from original. The authors should compare against some relevant literatures listed below and more.

The nature of the work makes it difficult to reproduce by others. This is understandable.
However, the main problem of the experiments in this paper is that it only experiments on small scale networks which make it far from generalizable.
Also, its baselines are rather naive.

**Strength And Weaknesses:**

+ Works on an important problem of closing the gap between the ideal vs memristor-based neural execution.
+ Does a fine job explaining the problem

- No novelty.
- Experiments are only provided in small scale networks which do not seem representative.


**Summary Of The Paper:**

This paper aims to adapt the neural network training to the non-idealities imposed by the memristor crossbars. The paper proposes hardware-Restriction-Aware Training (HRAT) which takes into account the finite-resolution of DACs, process variation of memristor devices, etc.
For each of the non-idealities, the paper proposes a counterpart layer to account for the non-ideality to be considered in the weights during inference.
The paper provides experimentation on FC-4 and LeNet-5 on MNIST, VGG-16 on CIFAR to show that it can better mimic the realistic behavior of the hardware compared to naive training.

**Summary Of The Review:**

This paper aims to adapt the neural network training to the non-idealities imposed by the memristor crossbars. The paper proposes hardware-Restriction-Aware Training (HRAT) which takes into account the finite-resolution of DACs, process variation of memristor devices, etc. For each of the non-idealities, the paper proposes a counterpart layer to account for the non-ideality to be considered in the weights during inference. The paper provides experimentation on FC-4 and LeNet-5 on MNIST, VGG-16 on CIFAR to show that it can better mimic the realistic behavior of the hardware compared to naive training.

The paper definitely works on an important topic that aims to bridge the gap between the simulation and the real hardware implementation of MNNs. The research direction of the paper has the potential to enable a significantly more efficient neural execution.

Another point is that the paper does a fine job describing the problem. Considering the fact that many people in the ML community might not be familiar with the technological advances in the hardware, this paper can serve a good starting point.

However, the paper has several issues. First of all, the paper seems to be ignoring the huge body of works in the computer architecture and design automation community. The community has looked into various ways to mitigate the non-idealities in the hardware through similar approaches. The paper currently compares against a naive baseline. However, it should compare against some of the relevant works cited below. In fact, it seems that the paper also misses out on some relevant works including:
* Shafiee, Ali, et al. "ISAAC: A convolutional neural network accelerator with in-situ analog arithmetic in crossbars." ACM SIGARCH Computer Architecture News 44.3 (2016): 14-26.
* Ghodrati, Soroush, et al. "Mixed-Signal Charge-Domain Acceleration of Deep Neural Networks through Interleaved Bit-Partitioned Arithmetic." Proceedings of the ACM International Conference on Parallel Architectures and Compilation Techniques. 2020.

Even if we ignore the fact that the paper does not make any novel contributions, the paper only experiments on small scale networks which make it far from generalizable.

---

### Decision · Program_Chairs · 2023-01-20

**Decision:**

Reject

**Justification For Why Not Higher Score:**

A clear cut case: all reviewers suggested rejection, and no response was submitted.


**Justification For Why Not Lower Score:**

N/A

**Metareview: Summary, Strengths And Weaknesses:**

This paper suggests a method (based on BN-fusing) to decrease accuracy loss due to weight-variation in memristor-based neural networks. Though this paper tackles and explains an important problem, the reviewers raised many issues. A central and recurring issue was the lack of novelty and comparisons with previous works (and here is another work that might be relevant: "Robust processing-in-memory neural networks via noise-aware normalization"). Therefore, all reviewers suggested rejection, and no response was submitted.